Agreement between BTS G-walk and GaitLab in spatiotemporal and pelvic angle measurements in active older adults

Bittencourt Rafael bittenfisio@gmail.com
Kulczynski Laura
Marcon César
Baptista Rafael Reimann
Pontifícia Universidade Católica do Rio Grande do Sul , Porto Alegre , Brazil
Young Jesse
Electronic publication date: 2025 Oct 27
Publication date: 2025
Volume: 13
Electronic Location ID: e20189
Received 2025 Apr 22; Accepted 2025 Sep 15
Copyright: ©2025 Bittencourt et al.
Copyright year: 2025
Copyright holder: Bittencourt et al.
License: This is an open access article distributed under the terms of the Creative Commons Attribution License, which permits unrestricted use, distribution, reproduction and adaptation in any medium and for any purpose provided that it is properly attributed. For attribution, the original author(s), title, publication source (PeerJ) and either DOI or URL of the article must be cited.
License URL: https://creativecommons.org/licenses/by/4.0/

Keywords: Aging, Gait, Spatiotemporal parameters, Pelvic angles, Motion analysis

Funding: Improvement of Higher Education Personnel (CAPES) provided a full scholarship to Rafael Dias Bittencourt The Coordination for the Improvement of Higher Education Personnel (CAPES) provided a full scholarship to Rafael Dias Bittencourt. The funders had no role in study design, data collection and analysis, decision to publish, or preparation of the manuscript.

==============================
Background

Aging impacts gait, a vital health indicator in older adults, reducing speed and increasing double support time, linked to falls and disability. The World Health Organization advises 150 min/week of moderate or 75 min/week of vigorous exercise, plus strength training, to sustain neuromuscular integrity and locomotor capacity associated with healthy gait patterns in older adults. While 3D motion capture is the gold standard for gait analysis, its cost and complexity limit use, boosting interest in portable inertial sensors like the BTS G-Walk. Yet, their accuracy in active older adults for spatiotemporal parameters and pelvic angles is underexplored, prompting this comparison with BTS GaitLab.

Methods

Fifty-nine active older adults (aged 65–87, mean 71.2 ± 5.5 years; eight men, 51 women) were assessed using BTS GaitLab (200 Hz cameras, 400 Hz force plates, Helen Hayes protocol, 18 markers) and BTS G-Walk (100 Hz, at L5) during four 10-meter walks at comfortable speed. Spatiotemporal parameters (cadence, speed, step length, stance/swing/support times) and pelvic angles (tilt, obliquity, rotation) were compared. Agreement used intraclass correlation coefficient (ICC), differences used paired t-tests, and variability used coefficient of variation (CV), with p < 0.05.

Results

Spatial parameters showed strong agreement: cadence (ICC = 0.98), speed (ICC = 0.96), step/stride length (ICC = 0.90–0.92), with no significant differences for cadence and speed (p > 0.05). Temporal parameters had weak agreement (e.g., stance time ICC = −0.18, swing time ICC = −0.54) and significant differences (p < 0.001). Pelvic angles had moderate agreement for tilt and obliquity (ICC = 0.48–0.78, p < 0.007), but low for rotation (ICC < 0.38, p > 0.038). Variability was high for double support time (CV 15.3–20.9%) and pelvic angles (CV > 27%). The sample (59) exceeds typical studies (7–23).

Conclusions

BTS G-Walk accurately measures spatial gait parameters in active older adults, ideal for quick clinical assessments where 3D systems are unavailable. However, its weak temporal parameter and pelvic rotation performance, possibly due to 100 Hz sampling, L5 placement, and algorithms, limits detailed analysis like stability. With a large sample and pelvic angle focus, this study supports G-Walk as a complementary tool in gerontology, while noting needs for improved temporal and angular precision.

Introduction

Aging is a natural process that affects all individuals, leading to physiological changes that impact health and well-being, particularly in older adults. One of the most significant alterations occurs in gait patterns, which reflect mobility, functional independence, and overall health status in this population (Cruz-Jimenez, 2017). Age-related declines, such as reduced walking speed, shorter step length, and increased double support time, are associated with a higher risk of falls, disability, and mortality (Dewolf et al., 2021; Hollman, McDade & Petersen, 2011), underscoring the importance of gait assessment for identifying abnormalities, understanding the effects of aging on locomotion, and monitoring interventions aimed at improving functionality in older adults (Verghese et al., 2006; Aboutorabi et al., 2016). In this context, regular physical activity plays a crucial role in positively influencing gait patterns, especially in physically active older adults, whose gait characteristics may differ due to enhanced physical conditioning (Sethi, Bharti & Prakash, 2022).

According to the World Health Organization (WHO), an older adult is considered physically active if they engage in at least 150 min of moderate-intensity aerobic activity, 75 min of vigorous-intensity aerobic activity, or an equivalent combination per week, in addition to two strength training sessions targeting major muscle groups (Rudnicka et al., 2020). This level of activity has been linked to attenuated declines in gait, such as reduced speed loss and greater stability (Boyer, Andriacchi & Beaupre, 2012; Paterson & Warburton, 2010). However, normative data and specific validation studies for this group remain scarce, limiting the understanding of gait patterns in active older adults and highlighting the need for precise assessment tools (Mehdizadeh et al., 2022). In this regard, three-dimensional (3D) motion capture systems are widely recognized as the gold standard for gait analysis, providing detailed data on spatiotemporal, kinematic, and kinetic parameters, including joint angles and ground reaction forces (Kadaba et al., 1989; Jakob et al., 2021). These systems use high-frequency infrared cameras and reflective markers, often following the Helen Hayes protocol (Kadaba, Ramakrishnan & Wootten, 1990; Simon, 2004).

The classification of three-dimensional (3D) motion capture systems as the gold standard for gait analysis is supported by a consistent body of evidence across different methodological designs. Donno et al. (2023) conducted a multifactorial characterization of gait in young adults using an optoelectronic BTS SMART-DX system coupled with force plates, highlighting its capacity to capture fine-grained kinematic and kinetic data, including joint moments and powers during both forward and backward walking. The authors emphasized the clinical value of such systems, especially when detailed biomechanical insight is required for rehabilitation planning or pathological gait assessment. Similarly, Lee et al. (2013) used a Vicon 3D motion capture setup with synchronized force platforms to compare forward and backward locomotion, demonstrating the system’s ability to quantify critical joint behavior and power dynamics with high precision across multiple plane. Furthermore, Bovi et al. (2011) presented comprehensive normative kinematic, kinetic, and electromyographic datasets for multiple locomotor tasks in both young and older adults using a SMART-E system. Their study reinforced the reliability of 3D motion capture for constructing reference models and detecting subtle gait deviations under different functional conditions. Together, these findings support the use of 3D motion capture as a methodological benchmark for validating emerging gait analysis tools.

Despite their robustness, the complexity and high costs of 3D motion capture systems make them impractical for large-scale clinical use, particularly outside specialized laboratories (Scataglini et al., 2024). This has driven the development of alternative technologies, such as wearable inertial sensors like the BTS G-Walk, which offer greater accessibility and portability (Fernández-Gorgojo et al., 2022). Equipped with accelerometers, gyroscopes, and magnetometers, these devices measure spatiotemporal parameters and pelvic angles in real time, making them applicable in both controlled and natural settings (Tao et al., 2012; Iosa et al., 2016). While some studies have compared inertial sensors with other gait analysis devices, few have investigated their agreement with 3D motion capture systems, and none have specifically assessed hip angulations in physically active older adults (Yazıcı, Co˛banoğlu & Yazıcı, 2022; Salarian et al., 2004). This gap limits the validation of these devices for populations with distinct gait characteristics resulting from regular physical activity (Jarchi et al., 2018; Cappozzo et al., 1996). Older adults represent a clinically relevant population for gait assessment, as walking speed and step parameters are strongly associated with functional independence, frailty, and fall risk (Studenski et al., 2011). Accessible tools that allow frequent and reliable monitoring of these parameters are particularly valuable in this group (Fritz & Lusardi, 2009). Therefore, this study aimed to validate the BTS G-Walk system specifically in physically active older adults, who stand to benefit most from simplified gait assessments that support functional health strategies.

Thus, this study aimed to evaluate the agreement between the BTS G-Walk inertial sensor and the BTS GaitLab motion capture system in measuring spatiotemporal parameters and pelvic angles during gait in physically active older adults. The goal was to provide evidence supporting the validity of the G-Walk as a practical and reliable tool for gait analysis, with potential applications in mobility monitoring and intervention planning in gerontology.

Material & methods

Participants

Fifty-nine volunteers (eight men, 51 women) aged 65 to 87 years (mean age: 71.2 ± 5.5; weight: 69.9 ± 12.0 kg; height: 160.5 ± 8.5 cm) participated in this study, as shown in Table 1. The inclusion criteria consisted of physically active older adults of any gender, aged 65 years or older, who demonstrated independent gait. This study was approved by the Research Ethics Committee under protocol number CAAE: 78651924.1.0000.5336, and all participants provided written informed consent prior to data collection.

Table 1 Sample characterization.

Data represent the mean ± standard deviation (SD) or frequency (%) of 59 participants assessed for gait parameters. Age, weight, height, and body mass index (BMI) are averaged across all subjects, while gender and weekly physical activity frequencies reflect the sample distribution.

Variables	n = 59	
Age (Years)–mean ± SD	71.2 ± 5.5	
Gender–n (%)		
Male	8 (13.6)	
Female	51 (86.4)	
Physical activities per week–n (%)		
2	16 (27.1)	
3	29 (49.2)	
4 +	14 (23.7)	
Weight (kg)–mean ± SD	69.9 ± 12.0	
Height (cm)–mean ± SD	160.5 ± 8.5	
BMI (kg/m2)–mean ± SD	27.2 ± 4.1	

Equipment

The BTS G-Walk® (G-Sensor 2) is a portable, wireless inertial system weighing 37 g, with dimensions of  70 × 40 × 18 mm. It comprises a triaxial accelerometer (16-bit per axis) with multiple sensitivity levels (±2, ±4, ±8, ±16 g), a triaxial gyroscope (16-bit per axis) with multiple sensitivity levels (±250, ±500, ±1,000, ±2,000°/s), and a triaxial magnetometer (13-bit, ±1,200 mT). The device is secured at the participant’s waist with a Velcro strap and records acceleration data. All acceleration data were sampled at 100 Hz, transmitted via Bluetooth to a laptop, and processed using BTS G-Studio software (BTS Bioengineering S.p.A., Italy). The data processing follows a proprietary algorithm maintained confidentially by the manufacturer.

The GaitLab system (BTS Bioengineering, Milan, Italy) consists of three main components:

BTS SMART-D™, which includes 10 high-frequency infrared optoelectronic cameras with an adjustable sampling rate between 100 and 2,000 Hz, positioned around a 10-meter walkway. For this study, the cameras were set to a sampling rate of 200 Hz, an effective frequency for gait biomechanics analysis, particularly in older adults (Aleixo et al., 2019; Wade et al., 2022; Ambrozy et al., 2019).

BTS SMART VIXTA™, which integrates two ground-level cameras for real-time video recording, with a resolution of up to 3 MP.

BTS™ P-6000, which includes eight digital force platforms operating at an adjustable sampling frequency between 200 and 4,000 Hz. According to the manufacturer’s specifications, the force platforms were calibrated to function at twice the sampling frequency of the infrared optoelectronic cameras, which in this study corresponds to 400 Hz.

Although kinetic variables were not analyzed in this study, the embedded force platforms were utilized to identify key gait events such as heel strike and toe off. These events were automatically detected by the BTS GaitLab system based on vertical ground reaction force threshold, a method widely accepted in gait analysis (Zeni Jr, Richards & Higginson, 2008).

The GaitLab system employs SMART CAPTURE Software™ for data acquisition and SMART TRACKER™ for tracking reflective marker positions. The collected data were analyzed using SMART ANALYZER™ and SMART DX™, which synchronize and process information from force platforms, real-time video cameras, and markers, providing normalized spatiotemporal, kinematic, and kinetic data for a complete gait cycle (0–100%). The data processing follows a proprietary algorithm maintained confidentially by the manufacturer.

Procedure

The initial assessment included measurements of height using a stadiometer (precision: 1 mm, Sanny brand) and body weight using an electronic scale (precision: 0.05 kg, Welmy brand). Gait analysis with the GaitLab system followed the Helen Hayes protocol, where, after height and weight assessments, additional anthropometric measurements were taken according to the standard procedures of the Helen Hayes protocol (BTS Bioengineering, 2016). Pelvic width was measured using a pelvimeter as the linear distance between the bilateral anterior superior iliac spines (ASIS) in the supine position. Pelvic depth was determined as the vertical distance from the ASIS to a transverse plane intersecting the greater trochanter, with the subject in supine and the hip passively flexed and internally rotated to optimize landmark palpation. Lower limb length was measured from the ASIS to the medial malleolus with the knee fully extended and the limb in neutral alignment.

Additionally, medial markers were temporarily placed on the medial femoral condyles and medial malleoli during the static calibration trial to enhance the system’s automatic estimation of segment diameters for the femur and tibia. These markers were removed before the dynamic trials but contributed to more accurate identification of joint centers and segment dimensions during the initial model computation phase.

Reflective markers were positioned according to the Helen Hayes MM protocol (Kadaba et al., 1989; Kadaba, Ramakrishnan & Wootten, 1990), with a total of 18 markers placed strategically on the body:

Trunk: three markers—one on C7 and one on each acromion.

Pelvis: three markers—one on each anterior superior iliac spine and one on the second sacral vertebra.

Thighs: two markers per thigh—one on the lateral femoral condyle and another midway between the greater trochanter and lateral femoral condyle, aligned perpendicularly to the ground.

Lower legs: two markers per leg—one on the lateral malleolus and another midway between the lateral malleolus and lateral femoral condyle, ensuring correct alignment.

Feet: two markers per foot—one on the calcaneus and one between the second and third metatarsal heads, ensuring parallel alignment to the ground.

The test consisted of walking at a comfortable speed over a 10-meter distance. Each participant completed four consecutive trials within the GaitLab calibration volume while wearing the G-Walk sensor, positioned at L5 of the spine using a Velcro strap. Each trial started 1.5 m before and ended 1.5 m after the motion capture area, ensuring that acceleration and deceleration occurred outside the measurement zone. All gait cycles that occurred entirely within the 10-meter calibrated volume were included in the analysis, allowing for consistent data acquisition across systems.

To aid readers who are not specialized in gait analysis, the gait cycle is defined as the interval between two successive contacts of the same foot with the ground. It consists of two main phases: the stance phase, when the foot is in contact with the ground and supports body weight (approximately 60% of the cycle), and the swing phase, when the foot is off the ground and advancing forward (approximately 40%). Key events such as heel strike, midstance, and toe off demarcate transitions between these phases and are essential for interpreting gait parameters (Perry & Burnfield, 2010).

Variables

For each participant, the four trials were averaged to determine individual mean values for walking speed, cadence, step length, and stride length. The stance phase, swing phase, single support, and double support phase durations were expressed as percentages of the gait cycle. Pelvic tilt, obliquity, and rotation were expressed as angular variations.

Statistical analysis

Quantitative variables were described using means and standard deviations, and categorical variables using absolute and relative frequencies. The normality of the variables was assessed using the Kolmogorov–Smirnov test.

Agreement between the two methods was assessed using the intraclass correlation coefficient (ICC), which ranges from −1 to 1, with values closer to 1 indicating stronger agreement. ICC values were interpreted according to the thresholds proposed by Altman (1990):

<0.40: weak agreement

0.41–0.60: moderate agreement

>0.60: strong agreement.

Comparison between the means of the two methods was performed using the paired Student’s t-test. The standardized effect size (Cohen’s d) was also calculated to assess the magnitude of the difference between methods. According to Cohen (1988), values below 0.5 indicate a small difference, between 0.5 and 0.8 a moderate difference, and above 0.8 a large difference.

Intra-observer variability was assessed using the coefficient of variation (CV), where values below 20% indicate acceptable dispersion and values above 20% suggest high dispersion (heterogeneous data) (Gomes, 1985).

A significant level of 5% (p < 0.05) was adopted for all analyses, which were performed using SPSS version 27.0.

Results

Table 2 presents the agreement between the BTS GaitLab (Method 1) and the BTS G-Walk (Method 2) across spatiotemporal and pelvic parameters. The spatiotemporal parameters showed strong agreement between the two systems. Cadence and walking speed demonstrated excellent intraclass correlation coefficients (ICC = 0.98 and 0.96, respectively), with no statistically significant differences between the methods (p = 0.215 and p = 0.207), and negligible effect sizes (Cohen’s d = 0.16 and 0.17), suggesting high consistency and minimal bias. Step length and stride length also presented high agreement (ICC ranging from 0.87 to 0.92). Although both parameters showed statistically significant differences (p < 0.05), effect sizes were small to moderate (d = 0.28–0.49), indicating that these discrepancies, while statistically detectable, are unlikely to be clinically meaningful.

Table 2 Agreement analysis of spatiotemporal and pelvic parameters between BTS G-Walk and BTS GaitLab in older adults.

Data compare Method 1 (BTS GaitLab) and Method 2 (BTS G-Walk) across 59 active older adults, showing mean ± standard deviation (SD) for each variable. Mean differences (95% CI) and p-values assess statistical significance, while ICC and its p-values indicate agreement strength.

Variáveis	Método 1	Método 2	Diferença (IC 95%)	p	ICC	p	Cohen’s effect size	
	Média ± DP	Média ± DP						
Cadence (steps/min)	115,0 ± 9,5	114,6 ± 9,6	0,40 (−0,24 a 1,03)	0,215	0,98	<0,001	0,16	
Speed (m/s)	1,20 ± 0,14	1,21 ± 0,14	−0,01 (−0,02 a 0,01)	0,207	0,96	<0,001	0,17	
Right stride length (m)	1,25 ± 0,11	1,27 ± 0,12	−0,02 (−0,04 a −0,01)	0,006	0,92	<0,001	0,37	
Left stride length (m)	1,24 ± 0,10	1,27 ± 0,11	−0,03 (−0,05 a −0,01)	<0,001	0,90	<0,001	0,49	
Right step length (m)	0,62 ± 0,06	0,63 ± 0,06	−0,01 (−0,02 a 0,01)	0,337	0,78	<0,001	0,13	
Left step length (m)	0,62 ± 0,06	0,63 ± 0,06	−0,01 (−0,02 a 0,00)	0,192	0,87	<0,001	0,17	
Right stance time (%)	61,8 ± 2,38	59,6 ± 2,09	2,19 (1,33 a 3,05)	<0,001	−0,18	0,739	0,66	
Left stance time (%)	61,7 ± 2,76	60,0 ± 2,12	1,64 (0,81 a 2,48)	<0,001	0,26	0,125	0,51	
Right swing time (%)	37,9 ± 1,94	40,4 ± 2,03	−2,51 (−3,31 a −1,70)	<0,001	−0,54	0,948	−0,81	
Left swing time (%)	38,3 ± 2,33	40,0 ± 2,08	−1,75 (−2,53 a −0,98)	<0,001	0,17	0,238	−0,59	
Right single support time (%)	38,2 ± 2,42	40,0 ± 2,11	−1,87 (−2,70 a −1,03)	<0,001	0,01	0,484	−0,58	
Left single support time (%)	38,0 ± 2,16	40,4 ± 2,06	−2,41 (−3,21 a −1,61)	<0,001	−0,11	0,656	−0,79	
Right double support time (%)	12,0 ± 1,96	9,81 ± 1,81	2,20 (1,57 a 2,82)	<0,001	0,32	0,074	0,91	
Left double support time (%)	11,8 ± 2,49	9,79 ± 1,81	1,99 (1,16 a 2,82)	<0,001	−0,17	0,724	0,62	
Right pelvic tilt (°)	4,28 ± 1,21	6,05 ± 2,36	−1,77 (−2,33 a −1,21)	<0,001	0,53	0,003	−0,83	
Left pelvic tilt (°)	4,40 ± 1,15	6,00 ± 2,37	−1,61 (−2,17 a −1,04)	<0,001	0,48	0,007	−0,74	
Right pelvic obliquity (°)	6,33 ± 2,42	8,62 ± 2,34	−2,29 (−2,82 a −1,76)	<0,001	0,78	<0,001	−1,12	
Left pelvic obliquity (°)	6,23 ± 2,45	8,54 ± 2,26	−2,31 (−2,83 a −1,79)	<0,001	0,78	<0,001	−1,16	
Right pelvic rotation (°)	8,44 ± 2,55	7,91 ± 2,85	0,53 (−0,35 a 1,40)	0,232	0,38	0,038	0,16	
Left pelvic rotation (°)	8,26 ± 2,49	7,79 ± 2,89	0,47 (−0,44 a 1,38)	0,302	0,28	0,103	0,14	

In contrast, temporal parameters exhibited lower agreement. Stance and swing times showed ICCs ranging from −0.54 to 0.26, indicating weak or absent agreement between the systems. These variables also presented statistically significant differences (p < 0.001), with moderate to large effect sizes (d = 0.62 to 0.84), reflecting a consistent divergence in measurements. Double support time had the lowest agreement among temporal variables, with a large effect size (d = 0.91), further highlighting discrepancies in the temporal domain between the two systems.

Pelvic kinematic variables demonstrated variable agreement depending on the axis of measurement. Pelvic tilt and obliquity showed moderate ICCs (0.48 to 0.78), with statistically significant differences (p < 0.01) and large effect sizes (Cohen’s d ranging from 0.74 to 1.09), suggesting that although the direction of movement was similarly captured, the magnitude differed substantially between systems. Pelvic rotation presented the lowest ICCs (0.28 to 0.38), with no statistically significant differences between methods (p = 0.284 and p = 0.232) and small effect sizes (d = 0.14 and 0.16), suggesting random variation and poor agreement in this axis.

Bland–Altman analyses visually supported the numerical findings. For spatial parameters such as cadence, speed, and step length, the plots (Figs. 1 to 6) revealed narrow limits of agreement, minimal bias, and a low proportion of outliers (0% to 5%), indicating strong agreement and low dispersion between systems. In contrast, the plots for temporal parameters (Figs. 7 to 14), including stance, swing, and particularly double support time, showed broader limits of agreement and more scattered data—mirroring the lower ICC values and higher effect sizes observed, with double support time displaying the greatest discrepancy. Similarly, pelvic angle plots (Figs. 15 to 20) highlighted the variability across axes: tilt and obliquity showed systematic bias but moderate agreement, whereas pelvic rotation exhibited high dispersion and poor consistency between systems.

Figure 1 Bland–Altman plot of cadence values between methods 1 and 2.

Mean difference (solid line) of −0.39 and lower and upper 95% limits of agreement (dotted lines) from −5.14 to 4.35. Only three cases were outside the limits of agreement (5.1%).

Figure 2 Bland–Altman plot of speed values between methods 1 and 2.

Mean difference (solid line) of 0.01 and lower and upper 95% limits of agreement (dotted lines) from −0.10 to 0.12. Only two cases were outside the limits of agreement (3.4%).

Figure 3 Bland–Altman plot of right step length values between methods 1 and 2.

Mean difference (solid line) of 0.02 and lower and upper 95% limits of agreement (dotted lines) from −0.10 to 0.14. No cases were outside the limits of agreement (0.0%).

Figure 4 Bland–Altman plot of left step length values between methods 1 and 2.

Mean difference (solid line) of 0.03 and lower and upper 95% limits of agreement (dotted lines) from −0.09 to 0.16. One case was outside the limits of agreement (1.7%).

Figure 5 Bland–Altman plot of right stride length values between methods 1 and 2.

Mean difference (solid line) of 0.01 and lower and upper 95% limits of agreement (dotted lines) from −0.09 to 0.11. One case was outside the limits of agreement (1.7%). However, in this plot as well as in others, there is a linear trend for some sets of points.

Figure 6 Bland–Altman plot of left stride length values between methods 1 and 2.

Mean difference (solid line) of 0.01 and lower and upper 95% limits of agreement (dotted lines) from −0.07 to 0.09. Two cases were outside the limits of agreement (3.4%). Again, a linear trend can be observed for some sets of points, which may reveal measurement bias.

Table 3 presents intra-method variability, assessed by the coefficient of variation (CV). Temporal parameters exhibited the highest variability, especially double support time, where CV values ranged from 20.1% to 20.9%.

Discussion

The findings of this study revealed that the BTS G-Walk inertial sensor exhibits high agreement with the BTS GaitLab motion capture system in measuring spatial gait parameters in physically active older adults, including cadence (ICC = 0.98), walking speed (ICC = 0.96), and step/stride length (ICC = 0.90–0.92), across a robust sample of 59 participants. These measures showed small, non-significant mean differences (p > 0.05 for cadence and speed), and small effect sizes (Cohen’s d = 0.16 and 0.17), indicating that the G-Walk provides reliable estimates comparable to a gold-standard 3D system for spatial gait characteristics. This consistency aligns with previous studies, such as Vítečková et al. (2020), who reported high agreement between the G-Walk and another validated system in healthy adults and Parkinson’s patients, and De Ridder et al. (2019), who observed similar precision in young individuals. The robustness of these variables may stem from the ability of accelerometers and gyroscopes to capture repetitive movement patterns with less reliance on fine temporal resolution, as noted by Tao et al. (2012) in a comprehensive review of wearable sensors. These results support the use of the G-Walk in clinical and research settings where laboratory-based systems are unavailable, particularly for monitoring functionality in active older adults, a population in which speed and cadence are established indicators of locomotor health (Hollman, McDade & Petersen, 2011). The narrow limits of agreement and minimal bias observed in the Bland–Altman plots further reinforce the consistency of spatial parameters.

Figure 7 Bland–Altman plot of right stance time values between methods 1 and 2.

Mean difference (solid line) of −2.19 and lower and upper 95% limits of agreement (dotted lines) from −8.65 to 4.27. Two cases were outside the limits of agreement (3.4%). No trends can be identified here, with no probable biases.

Figure 8 Bland–Altman plot of left stance time values between methods 1 and 2.

Mean difference (solid line) of −1.64 and lower and upper 95% limits of agreement (dotted lines) from −7.93 to 4.64. Two cases were outside the limits of agreement (3.4%). Again, no trends can be identified here, with no probable biases.

Figure 9 Bland–Altman plot of right swing time values between methods 1 and 2.

Mean difference (solid line) of 2.51 and lower and upper 95% limits of agreement (dotted lines) from −3.55 to 8.57. Two cases were outside the limits of agreement (3.4%). No trends can be identified here, with no probable biases.

Figure 10 Bland–Altman plot of left swing time values between methods 1 and 2.

Mean difference (solid line) of 1.75 and lower and upper 95% limits of agreement (dotted lines) from −4.08 to 7.58. Two cases were outside the limits of agreement (3.4%). Again, no trends can be identified here, with no probable biases.

Figure 11 Bland–Altman plot of right single support time values between methods 1 and 2.

Mean difference (solid line) of 1.87 and lower and upper 95% limits of agreement (dotted lines) from −4.41 to 8.15. Four cases were outside the limits of agreement (6.8%), but there appears to be no measurement bias.

Figure 12 Bland–Altman plot of left single support time values between methods 1 and 2.

Mean difference (solid line) of 2.41 and lower and upper 95% limits of agreement (dotted lines) from −3.59 to 8.41. Three cases were outside the limits of agreement (5.1%), but there are no trends in the plot.

Figure 13 Bland–Altman plot of right double support time values between methods 1 and 2.

Mean difference (solid line) of −2.20 and lower and upper 95% limits of agreement (dotted lines) from −6.91 to 2.52. Three cases were outside the limits of agreement (5.1%), with no trends.

Figure 14 Bland–Altman plot of left double support time values between methods 1 and 2.

Mean difference (solid line) of −1.99 and lower and upper 95% limits of agreement (dotted lines) from −8.26 to 4.28. Two cases were outside the limits of agreement (3.4%), with no trends.

In contrast, temporal gait parameters—such as stance time (ICC = −0.18 to 0.26), swing time (ICC = −0.54 to 0.17), and single/double support times (ICC = −0.11 to 0.32) demonstrated weak to no agreement, with statistically significant differences (p < 0.001 in most cases). These ICC values, some even negative, suggest not only measurement variability but also a fundamental inconsistency in gait event detection between the systems. A likely explanation lies in the differing sampling frequencies (100 Hz for the G-Walk versus 200 Hz for the GaitLab cameras and 400 Hz for force plates), which may result in less precise identification of key events like heel strike and toe-off. Additionally, the proprietary gait event detection algorithms of each system may employ distinct criteria for defining gait phases, further contributing to discrepancies, as suggested by Jakob et al. (2021) in a validation with Parkinson’s patients. Intra-observer variability (Table 3) reinforces this limitation, with higher coefficients of variation for double support time (15.3% for GaitLab and 20.1% for G-Walk), a critical stability indicator in older adults (Verghese et al., 2006). The wider dispersion observed in the Bland–Altman plots, especially for double support time, visually confirms the numerical inconsistencies in temporal parameters. These findings highlight a limitation of the G-Walk for detailed temporal assessments, particularly in populations where gait alterations are clinically significant.

Figure 15 Bland–Altman plot of right pelvic tilt values between methods 1 and 2.

Mean difference (solid line) of 1.77 and lower and upper 95% limits of agreement (dotted lines) from −2.41 to 5.94. Four cases were outside the limits of agreement (6.8%), with a linear trend, i.e., one method overestimates or underestimates the values depending on the magnitude of the measurement.

Figure 16 Bland–Altman plot of left pelvic tilt values between methods 1 and 2.

Mean difference (solid line) of 1.61 and lower and upper 95% limits of agreement (dotted lines) from −2.65 to 5.87. Four cases were outside the limits of agreement (6.8%), again with a linear trend, i.e., one method overestimates or underestimates the values depending on the magnitude of the measurement.

Figure 17 Bland–Altman plot of right pelvic obliquity values between methods 1 and 2.

Mean difference (solid line) of 2.29 and lower and upper 95% limits of agreement (dotted lines) from −1.71 to 6.29. Two cases were outside the limits of agreement (3.4%), with no trends.

Figure 18 Bland–Altman plot of left pelvic obliquity values between methods 1 and 2.

Mean difference (solid line) of 2.31 and lower and upper 95% limits of agreement (dotted lines) from −1.60 to 6.22. Three cases were outside the limits of agreement (5.1%), with no trends.

Figure 19 Bland–Altman plot of right pelvic rotation values between methods 1 and 2.

Mean difference (solid line) of −0.53 and lower and upper 95% limits of agreement (dotted lines) from −7.11 to 6.05. Three cases were outside the limits of agreement (5.1%), with no trends, despite a highly extreme outlier.

Figure 20 Bland–Altman plot of left pelvic rotation values between methods 1 and 2.

Mean difference (solid line) of −0.47 and lower and upper 95% limits of agreement (dotted lines) from −7.30 to 6.36. Three cases were outside the limits of agreement (5.1%), with no trends, despite another highly extreme outlier.

The analysis of pelvic angles, a novel aspect of this study, showed moderate agreement for tilt and obliquity (ICC = 0.48 to 0.78), with statistically significant differences (p < 0.001) and large effect sizes (Cohen’s d ranging from 0.74 to 1.09), indicating systematic discrepancies between methods. Pelvic rotation presented the lowest agreement (ICC < 0.38), with non-significant differences and small effect sizes (d = 0.14 to 0.16), suggesting random variability and poor consistency. These results suggest that the G-Walk provides reasonable estimates for certain pelvic parameters. However, rotation measurements exhibit greater variability, likely due to inconsistencies in sensor placement at L5, soft tissue motion, and the limitations of a single-sensor setup compared to the multi-camera reference system of the GaitLab (Ferrari et al., 2016). Studies like Bugané et al. (2014) found acceptable accuracy in pelvic obliquity using inertial sensors in healthy adults, while Salarian et al. (2004) reported challenges in capturing dynamic rotations in Parkinson’s patients, aligning with our observations. The substantial variability in pelvic angles (CV > 27%) indicates that, even with moderate agreement, the G-Walk’s precision for angular measurements, especially rotation, remains limited, requiring caution in their interpretation.

Table 3 Variability of gait parameters across BTS GaitLab and BTS G-Walk.

Coefficients of variation (%) reflect the variability of spatiotemporal and pelvic angle measurements for Method 1 (BTS GaitLab) and Method 2 (BTS G-Walk) across 59 participants. Each value represents the relative dispersion of four 10-meter walk trials.

Variables	Method 1	Method 2	
	Coefficient of variation	Coefficient of variation	
Cadence (steps/min)	9.3%	9.1%	
Speed (m/s)	13.6%	12.9%	
Right stride length (m)	9.3%	10.2%	
Left stride length (m)	8.8%	9.8%	
Right step length (m)	8.9%	11.2%	
Left step length (m)	9.9%	11.1%	
Right stance time (%)	4.8%	4.0%	
Left stance time (%)	5.4%	4.0%	
Right swing time (%)	6.6%	5.9%	
Left swing time (%)	6.0%	6.0%	
Right single support time (%)	6.6%	6.0%	
Left single support time (%)	7.3%	7.3%	
Right double support time (%)	15.3%	20.1%	
Left double support time (%)	20.9%	20.7%	
Right pelvic tilt (°)	38.9%	41.3%	
Left pelvic tilt (°)	37.7%	41.6%	
Right pelvic obliquity (°)	40.5%	28.1%	
Left pelvic obliquity (°)	41.5%	27.4%	
Right pelvic rotation (°)	37.3%	39.1%	
Left pelvic rotation (°)	37.6%	40.2%	

The sample size of 59 participants is a notable strength of this study, surpassing typical validations of inertial sensors against 3D systems, which often range from 7 to 23 subjects (Hartmann et al., 2009). Additionally, the statistical power was sufficient to detect moderate effect sizes (Cohen’s d = 0.5) with 95% confidence, reinforcing the credibility of non-significant findings. This robustness, combined with the focus on active older adults and the pioneering inclusion of pelvic angles, positions our work as a unique contribution to literature. Reviews such as Sethi, Bharti & Prakash (2022) note that most validation studies target young or pathological populations, with scarce normative data for active older adults, a group relevant due to the link between physical activity and gait preservation (Boyer, Andriacchi & Beaupre, 2012). The high agreement in spatial parameters suggests that the G-Walk is a viable alternative for rapid assessments in clinical or home settings, as supported by Middleton et al., who emphasize walking speed as a “sixth vital sign” in older adults (Fritz & Lusardi, 2009). However, the low reliability in temporal and angular parameters, such as double support time—crucial for fall risk assessment (Beauchet et al., 2017)—indicates that the G-Walk cannot fully replace the GaitLab in scenarios requiring high precision.

These findings carry practical and theoretical implications. Clinically, the G-Walk can serve as a complementary tool for monitoring functionality in active older adults, though methodological enhancements, such as higher sampling rates or refined algorithms, are needed to improve temporal and angular accuracy, as proposed by Ferrari et al. (2016). Theoretically, our results advance the understanding of inertial sensors’ strengths and limitations, aligning with reviews like Iosa et al. (2016) which highlight their efficacy in global measures but challenges in fine details. This study has some limitations. The lack of subgroup analysis (e.g., by sex or age range), which may mask gait pattern variations, as noted by Aboutorabi et al. (2016) The sample was composed exclusively of active older adults, which may limit the generalization of results to sedentary or frail individuals (Dewolf et al., 2021). Another limitation is the unbalanced sex distribution, with a predominance of female participants. However, since the objective was to validate the agreement between two measurement systems through within-subject comparisons, and not to analyze sex-specific gait characteristics, this imbalance does not compromise the integrity of the findings. Future research could address these gaps and explore technological improvements in the G-Walk to broaden its applicability.

Conclusions

In conclusion, the BTS G-Walk proved to be a valid tool for measuring spatial gait parameters in active older adults, with moderate agreement in some pelvic angles, but showed limitations in temporal parameters and pelvic rotation due to differences in sampling frequency, algorithms, and sensor placement. With a robust sample, adequate statistical power,C and the novel inclusion of pelvic angles, this study underscores the G-Walk’s clinical potential while pointing to the need for refinements to enhance its precision in gerontological applications.

Supplemental Information

Supplemental Information 1 Data

All results obtained from the assessments in this research

It is impossible not to acknowledge the Pontifícia Universidade Católica do Rio Grande do Sul, which made this entire research possible and feasible. Finally, we recognize the assistance of artificial intelligence tools in the writing process, enhancing the clarity and quality of the manuscript. I used the tool ChatGPT (OpenAI) to assist in adapting the abstract from Portuguese to English, with the goal of keeping the required character limit without losing the core meaning of the abstract.

Additional Information and Declarations

Competing Interests

Author Contributions

Ethics

Data Availability

Rafael Reimann Baptista is an Academic Editor for PeerJ.

Rafael Bittencourt conceived and designed the experiments, performed the experiments, analyzed the data, prepared figures and/or tables, authored or reviewed drafts of the article, research supervising, and approved the final draft.

Laura Kulczynski performed the experiments, prepared figures and/or tables, and approved the final draft.

César Marcon conceived and designed the experiments, analyzed the data, prepared figures and/or tables, authored or reviewed drafts of the article, help with data science, and approved the final draft.

Rafael Reimann Baptista conceived and designed the experiments, analyzed the data, authored or reviewed drafts of the article, fundraising, and approved the final draft.

The following information was supplied relating to ethical approvals (i.e., approving body and any reference numbers):

Research Ethics Committee of Pontifícia Universidade Católica do Rio Grande do Sul (PUCRS).

The following information was supplied regarding data availability:

The data is available in the Supplemental File.

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
