# Peer review of "Agreement between BTS G-walk and GaitLab in spatiotemporal and pelvic angle measurements in active older adults"

_PeerJ, doi:10.7717/peerj.20189_

## Round 0.1 · original submission · Major Revisions

This paper was reviewed by two peer reviewers and generally well received. Most suggested changes are relatively minor. However, I'm indicating major revisions given Reviewer 2's concerns about the statistical methods. It is always possible that a change in statistical analysis will lead to a reinterpretation of the data (which would be a major change to the paper).

Reviewer 1 ·

Basic reporting

Clear English is used throughout.

Literature references could be improved in support of the gold standard system used for the study.

Tables are clear and useful for representing the results of validation.

Experimental design

The research question is well-defined, but it should be better supported.

The investigation and experimental design appear to match high technical & ethical standards (Informed consent is cited in the paper).

Methods could be described in further detail.

Validity of the findings

Conclusions are clearly stated and coherent with both the research question and the results presented in the manuscript.

Additional comments

This paper tries to quantify the accuracy of the inertial sensor GWalk in measuring spatial and temporal parameters and pelvic angles during gait in older adults, with respect to the BTS GaitLab. The research appears well conducted; I have just a few comments:

Abstract
-In the background, please rephrase “to maintain gait”. I think that in a scientific paper, it could be explained more precisely.

Introduction
-I totally agree with the authors that this research could be useful, mainly for clinical purposes. However, the authors should better explain the chosen research question: why the BTS GWalk performs better or worse for older adults than for young adults?

-Lines 102-103: A sentence affirming that a certain technology is WIDELY recognized as a gold standard would require further references. For example, you may read these research articles, which provide proof of the 3D motion capture systems for a quantification of both “kinematic and kinetic parameters” (as properly reported by the authors): https://doi.org/10.3390/s23104671, https://doi.org/10.1016/j.gaitpost.2010.08.009, https://doi.org/10.1016/j.gaitpost.2013.02.014

Materials and Methods
-The sample seems very unbalanced, comprising just 8 men against 51 women. I understand the difficulty in find participants of that specific range of age; however, the authors should provide (maybe in the Discussion) an explanation of the possible effects of this choice on the results obtained.

-Please for each anthropometric parameter measured, specify the anatomical points considered for measuring the distance. For example, pelvic depth was measured as the distance between the great trochanter and the ipsilateral anterior superior iliac spine (I guess).

-Were medial markers used? In general, in this protocol, markers on medial malleoli and medial femoral condyles are useful to provide the system and automatic assessment of knee and ankle diameters.

-The reference number 26 could be placed more precisely along the manuscript text, from my point of view.

-It is not clear to me the role of the force platform in this study. Could you better clarify why they were included in the equipment?

-Please, provide a brief description of the phases of the gait cycles. It would be very useful for readers not specialized in gait analysis.

This research provides practical implications mainly for clinical settings, the results are clearly presented, and the discussion appears easy to read and coherent with the reported results.

·

Basic reporting

The paper used clear, unambiguous, professional English language throughout.

The paper included enough context and well-cited references to understand the purpose. References towards physical activity are unclear as they pertain to the scope of the study – their inclusion implies that physical activity will be a covariate or consideration in the agreement model, which they were not, so not entirely sure why this section is needed. The discussion references this as well. A lingering question remains regarding whether the study happened to only recruit “active” older adults or whether this was an additional inclusion criterion.

Yes, the paper structure was clear.

While no figures were included, the Tables were well formatted.

Yes, raw data was provided.

Overall the basic reporting is good. However, the relevance of physical activity needs further justification as it was not used in the statistical methods other than to describe the population.

Experimental design

Yes, the scope of the paper matches the scope of the journal.

Yes, the authors have a well-defined research question that is relevant and meaningful. The authors identify the need for accurate gait parameter data capture outside a clinical gait laboratory and want to determine the validity of one such system.

The authors argue the inclusion of 59 participants goes beyond standard practice, and I applaud the recruitment effort in the attempt to perform a rigorous investigation.

Yes. The sampling rates of the 3D cameras (control) compared to the comparison system are appreciated. However, it would be nice to know how many gait cycles per trial were used for the analysis and how they were selected (e.g. only trials with a force plate strike or the second full stride).

Validity of the findings

The paper established the benefit of the literature.

Data has been provided. Issues with a skewed sample heavily towards women (51 vs 8) were discussed as a limitation. Perhaps a subgroup analysis of the women would be of interest.

The point of including physical activity data is unclear, as it is not used for prediction or validation modeling.

Some issues with the statistical methods were found. The Student’s paired t-test is not sufficient to establish agreement, as not finding a significant difference is not equivalent to finding no difference. Furthermore, preliminary tests for the assumptions underlying the Student’s t-test should be performed, e.g. Kruskal-Wallis test for normal distribution of continuous variables. Regardless, a non-inferiority test would be more appropriate to establish agreement and Bland-Altman plots would be a welcome inclusion. For more information, please see: Ranganathan P, Pramesh CS, Aggarwal R. Common pitfalls in statistical analysis: Measures of agreement. Perspect Clin Res. 2017 Oct-Dec;8(4):187-191. doi: 10.4103/picr.PICR_123_17., Schumi J, Wittes JT. Through the looking glass: understanding non-inferiority. Trials. 2011 May 3;12:106. doi: 10.1186/1745-6215-12-106. PMID: 21539749; PMCID: PMC3113981., Walker J. Non-inferiority statistics and equivalence studies. BJA Educ. 2019 Aug;19(8):267-271. doi: 10.1016/j.bjae.2019.03.004. Epub 2019 Apr 24. PMID: 33456901; PMCID: PMC7808096. This forms the basis of requiring major revision for publication.

Yes, the paper’s conclusions draw on and are supported by the results.

---

## Round 0.2 · accepted · Accept

The authors have done a laudable job responding to the prior critiques. Both reviewers agree that this manuscript is ready to accept, and I'm happy to recommend this work for publication.

Reviewer 1 ·

Basic reporting

The authors provided clear answers to all my comments and implemented all the suggested modifications in the current version of the manuscript. No further comments.

Experimental design

The authors provided clear answers to all my comments and implemented all the suggested modifications in the current version of the manuscript. No further comments.

Validity of the findings

The authors provided clear answers to all my comments and implemented all the suggested modifications in the current version of the manuscript. No further comments.

·

Basic reporting

Clear, unambiguous, professional English language used throughout. The paper used clear, unambiguous, professional English language throughout.
Intro & background to show context. Literature well referenced & relevant. The paper included enough context and well-cited references to understand purpose. Revised submission has addressed physical activity as an inclusion criteria for the study population.
Structure conforms to PeerJ standards, discipline norm, or improved for clarity. Yes, the paper structure was clear.
Figures are relevant, high quality, well labelled & described. While no figures were included, the Tables were well formatted.
Raw data supplied (see PeerJ policy). Yes, raw data was provided.

Experimental design

Original primary research within Scope of the journal. Yes, the scope of the paper matches the scope of the journal.
Research question well defined, relevant & meaningful. It is stated how the research fills an identified knowledge gap. Yes, the authors identify the need for accurate gait parameter data capture outside a clinical gait laboratory and want to determine the validity of one such system.
Rigorous investigation performed to a high technical & ethical standard. Authors argue the inclusion of 59 participants goes beyond standard practice, and I applaud the recruitment effort in the attempt to perform a rigorous investigation.
Methods described with sufficient detail & information to replicate. Yes. The sampling rates of the 3D cameras (control) compared to the comparison system are appreciated. The added detail regarding the length of the walking pathway helps to establish reproducibility.

Validity of the findings

All underlying data have been provided; they are robust, statistically sound, & controlled. Data has been provided. Issues with skewed sample heavily towards women (51 vs 8), this was discussed as a limitation. The revised submission has made a good argument against drawing conclusions from subgroup analyses with a small sample size.
The revised submission has sufficiently addressed concerns regarding similarity testing, and the inclusion of the complete set of Bland-Altman plots was much appreciated. The combination of ICC and Cohen’s d effect size have illustrated the similarity where observed and showed which differences would not be clinically significant.
Conclusions are well stated, linked to original research question & limited to supporting results. Yes, the paper’s conclusions draw on and are supported by the results.